# A smart literature exploration environment for COVID-19 literature

## Abstract

Historically the vast amount of knowledge that experts publish has been increasing in such a pace that keeping up to date and having a full perspective, even in particular topics, has become quite challenging. Such is the case of the current COVID-19 pandemic were there are so many clinical notes, experiments, expert observations around the world that doctors, researchers, and public authorities struggle to explore pieces of related but not explicitly connected knowledge concerning to their respective duties.

To simplify the process of exploration of the literature related to COVID-19, we propose a smart literature analysis environment, which includes several NLP-powered components to enable a more efficient reading process. In particular, we propose a semantically-guided transversal reading. We believe that this type of reading can significantly benefit the process of grasping the prominent opinion and state-of-the-art of a particular aspect. Our strategy to provide this feature was to interlink all semantically related sentences by semantic-textual-similarity (STS).

Besides, we enrich the literature with named-entity recognition and disambiguation (NERD), using the major life science databases as entity sources, enable named-entity searches, provide network-graphs of the most interconnected publications and, an interactive tool to highlight the most central statements within an article. All these capabilities are embedded in an easy to use web environment[1].

## 1 Introduction

The current pandemic took the world by surprise in all aspects. Among several difficulties, it also brought to the fore a problem that has become increasingly urgent over the years: That the knowledge we are producing is stored and shared in a manner that is not in-line with the pace and accessibility that is needed.

We have witnessed an enormous effort and solidarity that all kinds of involved actors around the world had put in reporting and sharing their experiences and findings. However, these efforts have been often undermined by the lack of easy ways to find a particular piece of knowledge or, if found, to weigh the degree of support and consensus it has.

One way to help tackle the emergency is to provide users with assisted means to explore the literature.

The goal of the system described in this paper is to facilitate the identification of central items of knowledge within a collection of publications. We addressed it by taking some insights of the reading comprehension analysis from the psycholinguistic point of view which led us to propose a combination of Natural Language Processing (NLP) tools that respond to 4 main objectives: provide a shallow but fast view of how publications are related through a network of semantic connections, facilitate skim reading by highlighting the more central sentences in an article, make more sense of a publication set by enabling browsing across semantically related statements and, enrich the reader context by Named-Entity Recognition and Disambiguation.

We developed these NLP strategies and integrated them in a web environment to make them easily available to any user. The application is already online and we are currently processing the firsts collections.

## 2 Background

To grasp the nitty-gritty of a document or a collection of them is clearly much more than summariz-

---

[1] http://covid19.ccg.unam.mx:82/

ing or finding interconnections but, comprehend what is expressed in the texts is part of the process. There is a long and solid research about the reading-comprehension phenomena which includes a myriad of sub-tasks. Among these sub-tasks it is to identify the topic, to make sense of the way information is organized and, to extract the main idea (Baumann, 1984).

We think that there are some interesting similarities between trying to get the main-idea of a single text and of a collection of them. Two key factors are the identification of words relevant in the context and the recognition of the relationships between the text propositions (Graesser et al., 1994; Grabe, 2004). The first one give important clues to what the main information is (Wilawan, 2012; Hoey, 1991). The second help to understand the way in which information is organized (Grabe, 2004; Crosson and Lesaux, 2013).

Automatic signaling these factors can play a potential support for the reader to enhance his comprehension (Degand and Sanders, 2002). Even though going beyond the mere identification is appealing, we think that doing that could misguide, in a restrictive way, the reader comprehension. This has been argued by some constructionist theories that postulate that during reading the meaning representations are generated online and that the readers generate representation that address their personal goals (Graesser et al., 1994). Therefore highlight relevant terms and signalize the relationships is a good trade-off between automatic-support and freedom.

## 3 Methodology

In this section, we first briefly describe the general pipeline and system architecture, and then the NLP methods and how they are leveraged.

### 3.1 System architecture and pipeline

First the publication collection is processed by our PDF-content-extractor which produces a set of files with text and stylographic data. These files are provided as input to the Semantic Textual Similarity (STS) and Named-Entity Recognition and Disambiguation (NERD) tasks which are performed individually and off-line. The publications' text and the annotated entities, resulting from the NERD step, are indexed in an ElasticSearch instance. The STS scores and its involved sentences indexes are stored in a key-value database (SSDB[2]). Finally, the collection name and metadata is registered in a MongoDB instance.

The front-end is composed by an Angular application and a Java API which access the above mentioned databases; all deployed as docker containers.

### 3.2 Semantic Textual Similarity

To evaluate all semantic similarities within a set of documents the STS has to be computed for all combinations of each sentence versus all the others, i.e., $\binom{N}{2} - N$ where N is the number of all the sentences in the set; which means an upper bound complexity of $O(N^2)$. So we opted for computing embeddings individually for all sentences and, after, measure the cosine between each pair of embeddings.

We used SciBERT (Beltagy et al., 2019), an unsupervised transformer language model pre-trained in the scientific literature. It was selected over BioBERT (Beltagy et al., 2019) because COVID-19 corpora include literature from other areas besides biological sciences, e.g., social sciences, public health, psychology, etc. Using SciBERT we mapped tokens to embeddings and then applied mean pooling to get one fixed-sized sentence vector. Due to the lack of STS corpora specific to the COVID-19 literature, we did not apply any fine-tuning.

Although we did not have an STS corpus for COVID-19 literature, we performed an indirect validation over one corpus of the topic of transcriptional-regulation (Lithgow-Serrano et al., 2019) (i.e., within the genetics field). We compared SciBERT versus DistilBERT (Sanh et al., 2019) and two versions of InferSent (Conneau et al., 2017) embeddings generated from different pre-trained GLOVE word-embeddings (Pennington et al., 2014), the results are shown in table 1.

| Pearson | Spearman | Model |
|---------|----------|-------|
| 0.686 | 0.743 | SciBERT |
| 0.645 | 0.620 | DistilBERT |
| 0.534 | 0.584 | InferSent GloVe-840B |
| 0.469 | 0.574 | InferSent GloVe SMTR |

Table 1: Model correlations

---

[2]http://ssdb.io/

### 3.2.1 Interlinking

The interlinks of each sentence (anchor) are the $N$ more semantically similar sentences within the publication set. Those target sentences can be in the same publication as the anchor (internal) or in others publications (external). We also filter-out those interlinks with a STS below a threshold $\theta = 0.85$.

It is worth noticing that if $s_j$ is within the best connections of $s_i$ ($s_i \rightarrow s_j$) does not necessarily hold the other way around ($s_j \rightarrow s_i$).

### 3.2.2 Network of publications

The network of publications is a directed graph with nodes representing publications and edges the semantic connections among them. There are two links between two publications $(A, B)$: the edge that summarize (eq. 1) all sentences from $A$ that target sentences in $B$ and, the edge that represent sentences from B targeting sentences in $A$.

$$edge(A \rightarrow B) = \sum^{|s_i \in A \wedge s_j \in B \wedge s_i \rightarrow s_j|} STS(s_i, s_j) \quad (1)$$

### 3.2.3 Highlight of more central sentences

The identification of the more central sentences in a publication is also based in the STS scores. For this task we only use the internal interlinks i.e., links connecting sentences in the same document. The centrality of a sentence $i$ within a document $A$ is given by the weighted sum of the connections of $s_i$ to other sentences plus the connections from other sentences to $s_i$ (eq. 2). The current weights were empirically set at $\psi = 0.35$ and $\omega = 0.65$, i.e., to evaluate the centrality of a sentence we give more importance to the connections it receives.

$$\Xi(s_i) = \sum^{|sentences_A|} \psi STS(s_i \rightarrow s_j) + \omega STS(s_j \rightarrow s_i) \quad (2)$$

### 3.3 Named-entity recognition and disambiguation

The NERD capabilities are provided by the OntoGene's Biomedical Entity Recogniser (OGER) (Furrer et al., 2019), a state-of-the-art biomedical annotator which in turn depends on the Bio Term Hub (BTH) (Ellendorff et al., 2015). BTH is a combined terminological resource created by dynamically sourcing entity names and their identifiers from reference databases.

OGER was integrated using its RESTful web service that allows remotely batch annotate a collection of documents. We opted for the TSV output which was later converted to the ElasticSearch annotation format.

## 4 Results

Figure 1 shows the reading environment. From right to left: first the toolbar allows easy access to the features that can be used when reading a publication; next the tool-details panel contains information and controls specific to the active tool; following, the PDF reader display the source PDF when available; at the middle there is the eagle-view panel that shows a global perspective of the document and where the highlighted sentences are located; finally, the processed content panel display the publication sentences (one per line), the annotated entities (if the annotation tool is active) and the hyperlinks to access interlinked sentences.

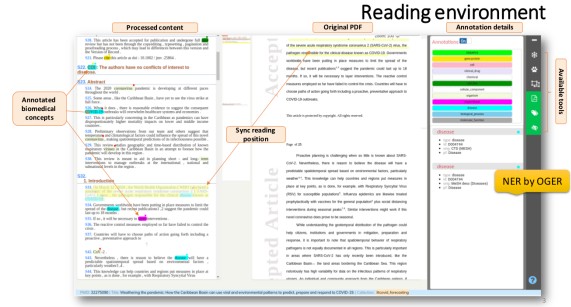

Figure 1: Reading component with highlighted named-entities.

The index of each sentence in the content-panel is a hyperlink that trigger the *interlink-tool* (fig. 2). If activated, the tool-detail panel display a list of internal and external sentences semantically related to the selected one. The list of internal relations works as a fast overview of the progression of the idea through the document and, the list of external relations is a summary of potential supporting statements over the publications collection. If the user want to inspect any of the target sentences in their respective contexts, he/she can click on the interlink and the content-panel would be updated to show the new content and focus the target sentence. In this way, the user can continue his reading across different publications (of the same collection) chasing the statements that he is interested in.

When the *highlighter-tool* is active (fig. 4) the

Browse articles through semantically similar sentences

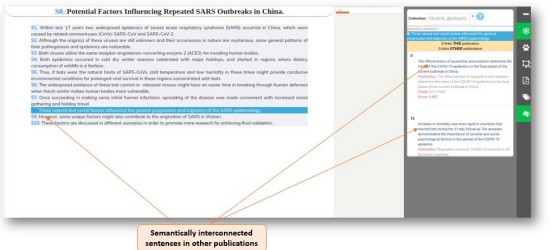

Figure 2: Interlink feature.

tool-detail panel shows a slider that controls the percentage of highlighted sentences. If, for example, the slider is set at 25%, the publication sentences would be ranked by their centrality and only the top 25% would be highlighted.

Extractive summarization

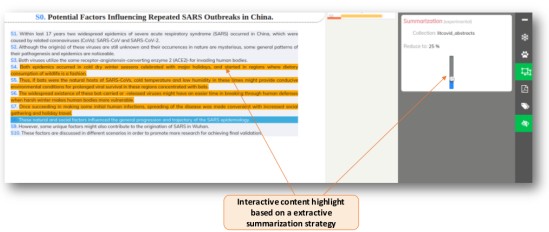

Figure 3: Highlighter based on sentences' centrality.

The system has been designed to handle multiple collections of publications. It is worth noticing that a document can belong to more than one collection and even though the its content remains the same, the generated interlinks depend on the other documents in the collection.

This application has also the capability to generate a network showing how the publications are interlinked within a collection (fig. 1). In this graph the width and the color of the edges are proportional to the interlinks strength so, it can be used as a first approach to graphically inspect which publications are more connected with others or which ones are isolated (e.g., could be due to divergent or contradictory statements). The graph is interactive and when the nodes are selected the immediate edges and connections are emphasized and the publication meta-data is shown in a box above the network.

A search module that leverages the ElasticSearch-features was included. To facilitate user interaction, we developed a very basic meta-language that enables them to search text

Searches on content and metadata

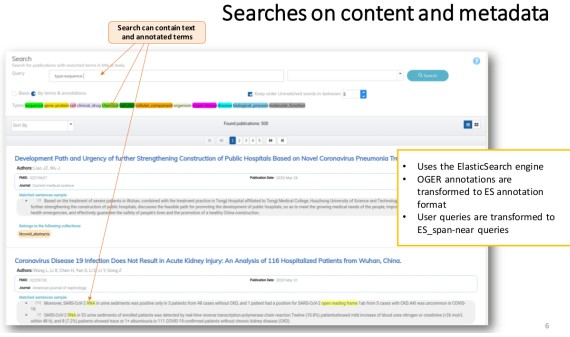

Figure 4: Interlinked-network of a publications.

and annotated entities in the same query. For example, the query "type:clinical_drug | recovery" would search in publications' titles and content for sentences that mention a drug (supposing that OGER identified it) and the word *recovery*.

One important aspect of the application is its usability. This could be affected by the computation time of the NLP tasks, so we took the decision to compute them off-line. Each collection of articles is processed once by each NLP tool and the results are distributed in the system's databases. Thus, when users interact with the web interface this only access previously stored content through an API.

## 5 Conclusions and future work

Drawing on some solid insights from existing research on reading-comprehension, we have developed a system that uses NLP methodologies implemented with state-of-the-art approaches to explore the COVID-19 literature in novelty ways.

We relied on two NLP methods: Semantic Textual Similarity (STS) and Named-Entity Recognition and Disambiguation (NERD).

STS was implemented using SciBERT, applying mean pooling to get one fixed-sized sentence vectors and then using the cosine measure. We did not apply any fine-tuning.

The NERD depends on OGER, a state-of-the-art biomedical entity recogniser that interoperates with the terminological aggregator Bio-Term Hub.

The outputs of these methods were leveraged in the following implemented tools: networks of semantically connected publications, highlighting of central sentences, NER tagging on publications' content, full-text and named-entities search and, transversal reading by following STS-based links.

Future work would focus on integrating techniques of discourse structure analysis.

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
