# OpenReview forum: "A smart literature exploration environment for COVID-19 literature"
_EMNLP/2020/Workshop/NLP-COVID — Submitted to NLP-COVID19-EMNLP_

### Official Review · AnonReviewer1 · 2020-09-23
**An application for COVID-19 literature exploration**

**Rating:** 4
**Confidence:** 4

**Review:**

The study authors describe an NLP platform that aims to identify central knowledge items within a collection of publications by: 1) highlighting the central sentences in an article; 2) creating a semantic connection graphs of related articles 3) enabling browsing across semantically related statements an; 4) and enabling named entity searches. To achieve these tasks, the authors use Semantic Textual Similarity (STS) and Named-Entity Recognition and Disambiguation.

While the concept is interesting, the authors do not demonstrate its accuracy or conduct any evaluations of the platform on COVID-19 literature, but rather perform an indirect validation of STS on the topic of transcriptional regulation. In the results section, the study authors primarily describe the features of the tool.

From the information that was presented in the paper, it is difficult to determine the applicability or the effectiveness of the platform in aiding knowledge identification/discovery/exploration and summarization as it relates to a diverse and rapidly expanding evidence base. In building the application, the authors use exising NLP platforms, namely SciBERT and OGER, a biomedical/genetic entity recognizer. In order to better assess the publication, the authors would need to provide further examples of the tool's performance.

---

### Official Review · AnonReviewer2 · 2020-09-24
**Promising approach, but evaluation lacking**

**Rating:** 5
**Confidence:** 4

**Review:**

This demo paper presents a system for searching COVID-19 literature and relating similar sentences both within and across documents. The search component is augmented with concept search. The goal according to the authors is to facilitate skim reading by visualising  salient connections between sentences.

I would like to applaud the authors for this effort which shows that a lot of work has been put into making the system operational. I find the eagle-view of the internal and external links interesting and definitely worthy of further exploration.

I would also like to raise several shortcomings, possible misunderstandings from my part, and suggestions for improvement.

In section 3.2, a corpus on transcriptional regulation is introduced, but little is said about it and the annotations it contains. Similarly, a short description of DistilBERT is in place, as well as motivating the decision to include this model in the present study.

I find interpreting the results from Table 1 quite difficult: How exactly was this study involving correlation carried out? What does e.g. the row including SciBERT tell us? Is it about correlation of cosine scores to human judgements in the transcriptional-regulation corpus?

In 3.2.2 (Network of publications), could the authors say more about the directionality of the graph? Is it not the case that edge weight e(A->B) should equal e(B->A) since the set of sentence connections is the same in both cases, and the cosine is symmetrical? Similarly, eq. (2) distinguishes between s_i->s_j and s_j->s_i, but I fail to see how these can be different.

If I wrongly understand STS to be the cosine similarity, please make that clear in the paper.

The coverage of the tool appears rather limited; how will the tool scale to larger collections?

For computing semantic similarity over a large number of sentence(/document) combinations, Maximum Inner Product Search (MIPS) algorithms may be relevant. These are used to find the approximate top k documents, using running time and storage space that scale sub-linearly with the number of documents.

There is little evaluation in the current paper, both in terms of the appropriateness of the STS measure and the usability, and no related tools are discussed either. This paper contains a substantial related works section, which may be useful: https://arxiv.org/pdf/2008.07880.pdf.

I would suggest the authors to not use “smart” to describe the tool as it doesn’t tell the reader much, and there’s no explanation in the paper what makes the tool smart. Also, try to provide a more informative title for the paper, e.g. “exploring COVID-19 literature with semantic similarity networks”.

Finally, the paper should be proofread (e.g. were there → where there,
firsts → first, and many more), and checked for coherence, e.g. in the sentence:

“To grasp the nitty-gritty of a document or a collection of them is clearly much more than summarizing or finding interconnections but, comprehend what is expressed in the texts is part of the process.”

---

### Official Review · AnonReviewer3 · 2020-09-25
**an exploration tool for COVID-19 literature**

**Rating:** 5
**Confidence:** 4

**Review:**

This paper introduces an application that enables researchers to explore COVID-19 literature. The functions of the application include finding similarity between publications, targeting central sentences in a publication, and extracts NER from publications.

Pros:
* This application could be useful for researchers to conveniently explore publications related to COVID-19, or maybe any other topics not limited to COVID-19.
* The methodology is reasonable.

Cons:
* While the proposed methods are reasonable, the lack of direct evaluation raises the question how well they  perform. For example, STS seems a key component in constructing the publication network and finding central sentences, but there is no direct evaluation how well the proposed method finds similar publications. Despite the indirect evaluation using "one corpus of the topic of
transcriptional-regulation", the performance of SciBERT may not be consistent in STS corpus from other domain.  If the STS is already poor at the first place, errors could propagate and amplify in the pipeline and make results less accurate. Also in finding central sentences, how good is using STS by SciBERT compared to other traditional methods, such as traditional text ranking algorithms? And there is no direct evaluation of the OGER in COVID-19 publications. Could OGER be poor in recognizing COVID-19 NERs that have not been seen in the training data of OGER? Without any quantitative analysis to answer these questions, the application is less useful to the community.

Questions:
* How did the authors chose 0.35 and 0.65 for Eqn.2 ?

In general the application could be potentially promising and helpful, but the authors need to have some direct evaluation to show the efficacy of the application.